# Advances in BBB on Chip and Application for Studying Reversible Opening of Blood–Brain Barrier by Sonoporation

**DOI:** 10.3390/mi14010112

**Published:** 2022-12-30

**Authors:** Yicong Cai, Kexin Fan, Jiawei Lin, Lin Ma, Fenfang Li

**Affiliations:** 1Shenzhen Bay Laboratory, Institute of Biomedical Engineering, Shenzhen 518107, China; 2School of Physics and Optoelectronic Engineering, Guangdong University of Technology, Guangzhou 510006, China; 3School of Medicine, The Chinese University of Hong Kong, Shenzhen 518172, China

**Keywords:** blood–brain barrier, microfluidics, ultrasound-mediated drug delivery

## Abstract

The complex structure of the blood–brain barrier (BBB), which blocks nearly all large biomolecules, hinders drug delivery to the brain and drug assessment, thus decelerating drug development. Conventional in vitro models of BBB cannot mimic some crucial features of BBB in vivo including a shear stress environment and the interaction between different types of cells. There is a great demand for a new in vitro platform of BBB that can be used for drug delivery studies. Compared with in vivo models, an in vitro platform has the merits of low cost, shorter test period, and simplicity of operation. Microfluidic technology and microfabrication are good tools in rebuilding the BBB in vitro. During the past decade, great efforts have been made to improve BBB penetration for drug delivery using biochemical or physical stimuli. In particular, compared with other drug delivery strategies, sonoporation is more attractive due to its minimized systemic exposure, high efficiency, controllability, and reversible manner. BBB on chips (BOC) holds great promise when combined with sonoporation. More details and mechanisms such as trans-endothelial electrical resistance (TEER) measurements and dynamic opening of tight junctions can be figured out when using sonoporation stimulating BOC, which will be of great benefit for drug development. Herein, we discuss the recent advances in BOC and sonoporation for BBB disruption with this in vitro platform.

## 1. Introduction

As the population ages, central nervous system (CNS) diseases such as Alzheimer’s disease and other dementias have become a great challenge, costing USD 305 billion in the U.S. by 2020, and these numbers are likely to rise to USD 1.1 trillion by 2050. Additionally, by the mid-century, the number of aged Americans with Alzheimer’s disease may climb up to 13.8 million, twice as many as today [1]. 

However, drug development for CNS diseases is complex. The presence of the BBB hinders drug delivery to the brain. Only small molecules around 500 Da can pass the BBB, but most drugs are much larger [2]. Additionally, most of the conventional pre-clinical tools have significant drawbacks. For example, the Transwell model is considered a simple way to rebuild the BBB in vitro, which allows for the co-culture of different types of cells in the BBB [3]. Nevertheless, the Transwell model is over-simple and static. It lacks mechanical stimuli that are critical for BBB integration [4]. The other way to study the BBB is to use in vivo animal models. However, the results of pre-clinical assessments may significantly deviate from human trials, considering the large differences in the BBB between species such as the expression of P-glycoprotein (P-gp), multidrug resistance-associated proteins, transports, and claudins [5]. 

Therefore, there is an urgent need for a preclinical drug screening platform that is cost-effective, less time consuming, and has fewer ethical issues and deviates less compared to in vivo animal models. Additionally, the BOC, a new production led by microfabrication and microfluidics, is an attractive way to reconstruct the BBB in vitro due to its controllability, transparency, and ease in resistance measurement and permeability assessment. Additionally, it can provide a repeatable and stable environment for drug screening, which could accelerate drug development [6]. Notably, a valid in vitro model must copy the BBB’s vital features involving the physiological and biological characteristics, and the BOC is considered a potential solution [7]. 

Is an effective in vitro model the final answer? Certainly not. During the past decades, various methods for delivering drugs to the brain have been proposed including non-invasive and invasive methods, as shown in Table 1. Among these, chemical methods are non-invasive and have a long history for opening the BBB. However, this is not the best answer due to its limitations. For instance, chemical agents such as sugar alcohols, solvents, and vasodilators used for drug delivery to the brain can cause serious side effects [8]. A typical example is mannitol, which is used to increase intracranial pressure. It can temporarily destroy the BBB reversibly by the vasodilatation and shrinkage of endothelial cells (ECs) [9]. However, it also exposes the brain globally to drug molecules and other harmful compounds from the circulatory system [10]. 

Instead, physical means holds great promise for opening the BBB. It has been found that ultrasound driven microbubbles can generate transient pores on the cell membrane, as the result of either persistent and low-amplitude oscillations of microbubbles (stable cavitation) or sharp expansion and collapse of microbubbles (inertial cavitation). This process is called sonoporation. Sonoporation can reversibly disrupt the BBB and increase BBB permeability to allow small compounds to pass through. Sonoporation has been demonstrated to be well-controlled, of low toxicity, and can treat locally, rather than the entire brain [11]. Its safety and efficiency in animals and humans have been reported in several studies [12,13,14]. 

In this review, we will introduce different kinds of BOCs and the opening of the BBB through sonoporation. Two aspects will be covered: (1) in vitro models of BBB and characterization, and (2) recent advances on BBB opening using ultrasound in the platform of BOC.

**Table 1 micromachines-14-00112-t001:** Different methods of drug delivery to the brain.

Method	Advantages	Disadvantages	References
Intranasal drug delivery	Non-invasive; direct reach to the brain	Affected by the nature of nasal mucosa; low efficiency	[15,16,17]
Chemical agents (osmotic procedure)	Well-developed; reversible disruption of the BBB	Pre-conditions needed; the whole BBB is exposed; neurotoxins influx	[8,9,10,18]
Local delivery (surgery)	Highly effective; repeatable; minimized systemic exposure; controlled drug releases time	Invasive; limited drug concentration and distribution; infection risks	[19,20]
Sonoporation	Non-invasive; minimized systemic exposure; reversible opening; well controlled; highly effective	Equipment needed; neurotoxins influx;	[21,22,23]

## 2. Structure of BBB and Substance Transportation

### 2.1. Cellular Structure of BBB

The BBB is a selective physiological barrier that controls transport between the blood and the CNS to maintain homeostasis and the neural microenvironment for optimal brain function [24]. The BBB is composed of brain microvascular endothelial cells (BMVECs) that are surrounded by pericytes and astrocytes (Figure 1A). Astrocytes are involved in many biological processes such as the uptake and release of neurotransmitters and the response to stress [25,26]. For example, in the case of trauma or pathological tissue injuries, astrocytes are activated and express more glial fibrillary acidic protein and vimentin. Pericytes also play an important role by regulating blood flow [27]. Moreover, in the BBB, blood vessels are surrounded by the layer of BMVECs and its basement membrane (BM), bordered by epithelial meningeal cells and associated extracellular matrix (ECM) and outer astroglial basement membrane and astrocyte endfeet [28]. These vascular membranes contribute to the integrity of the blood–brain barrier, forming a three-dimensional protein network mainly composed of laminin, collagen IV isomer, nidogen, and heparan sulfate proteoglycan. These proteins mutually support interactions between brain capillary endothelial cells, pericytes, and astrocytes [29]. 

The BOC models have been developed to better simulate real BBB functions. Endothelial cells are a crucial part of the BBB simulation. Different types of endothelial cells have been tested for the construction of BOC. For example, human cardiac microvascular endothelial cells (hCMEC) are favored by many researchers because they are easy to obtain and culture. However, they face the problems of poor barrier leakproofness and low TEER value [30,31,32]. Primary human brain microvascular endothelial cells (hBMEC) isolated from fresh brains are also used, which exhibit a larger endothelial electrical resistance and a higher level of protein expression of the tight junction (TJ), resulting in a more compact structure of the BBB. However, the disadvantages are expensive and less available [33,34,35]. Human umbilical vein endothelial cells (HUVEC) isolated from the umbilical cord, despite their simplicity and accessibility, obtain progressively lower TEER values and loss of function with increasing passages [34,36,37]. Human induced pluripotent stem cell-brain microvascular endothelial cells (hiPSC-BMEC) are derived by inducing stem cell differentiation. Although they demonstrate high TEER values and good formation of TJs, their differentiation process is complex and the allowed passage number is low [38,39,40].

**Figure 1 micromachines-14-00112-f001:**
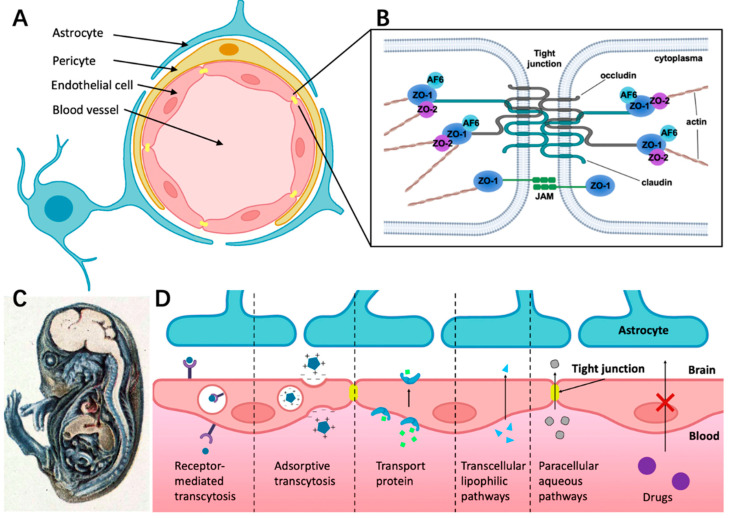
The structure of BBB and different transport pathways for substances across the BBB. (**A**) The BBB consists of endothelial cells (ECs) surrounded by pericyte and astrocyte endfeet. (**B**) The tight junction is composed of complex and diverse proteins including transmembrane proteins (claudins and occludins), zonula occludens proteins (ZO-I, ZO-II, and ZO-III), and junction-associated molecules (JAMs). (**C**) Demonstration of BBB hindering trypan blue entering the CNS in guinea pig embryo. The dye infected almost all of the tissue in the body except most of the brain, which indicates that the CNS is a closed compartment separated from the rest of the embryo [41]. (**D**) Schematic overview of specific transport pathways through the BBB; larger solutes can pass the BBB by receptor-mediated transcytosis and adsorptive transcytosis. Transport protein could cross the BBB through an active transport mechanism. The transcellular lipophilic pathway allows for lipid-soluble agents to pass the BBB. The paracellular pathway enhances the diffusion of water-soluble agents through tight junctions.

### 2.2. Structure Foundation of Tight Junctions (TJs)

Endothelial cells in the BBB have the main features of extremely low rate of transcytosis and forming restrictive paracellular diffusion barriers, namely tight junctions (TJs), which greatly limits the efficiency of substance transport [24,42]. 

TJ consists of transmembrane proteins including occludin, claudin, the cytoplasmic scaffold proteins ZO-1, -2, and -3, the actin cytoskeleton, and associated signaling proteins (Figure 1B) [43]. As shown in Figure 1B, ZO-1, -2, and -3 link junctional transmembrane proteins such as occludin and claudin to the actin cytoskeleton.

### 2.3. Different Ways for Substances Crossing the BBB under Physiological Conditions

It is well-known that the BBB has selective permeability. By injecting trypan blue systemically into guinea pig embryos, Wislocki et al. [41] observed that the dye stained nearly all of the body tissue, except for most of the brain and the spinal cord (Figure 1C). However, this was not the case when the dye was injected directly into the brain. A diverse combination of physical and chemical barriers constitutes the BBB, which hinders the flow of blood solutes into and out of the brain including drug delivery. Drug transportation across the BBB is complicated by factors such as molecular size, hydrophilicity, intercellular adhesiveness, and efflux transporters (e.g., P-gp) [44]. As a result of the poor drug transport into the brain, several clinical trials on CNS drugs for Alzheimer’s and Parkinson disease are hindered. 

Figure 1D presents different ways for substances crossing the BBB under physiological conditions. Receptor-mediated transcytosis can greatly enhance the efficiency of drug delivery through ligand binding to the membrane receptor of endothelial cells [44]. Adsorption-mediated transcytosis (AMT) provides another means of delivering drugs to the brain across the BBB. This method offers the potential to bind and uptake cationic molecules to the luminal surface of endothelial cells, followed by exocytosis at the abluminal surface [45]. Transporter protein such as glucose transporter isoform 1(GLUT-1) could cross the BBB through an active transport mechanism. For example, glucose first binds with the transporter proteins at the blood vessel side of the BBB, then the conformational change of transporter proteins helps transfer glucose or amino acids into the brain side [46,47]. The transcellular lipophilic pathway provides approximately 100 cm^2^/2 g of brain surface area per average human brain mass for drug delivery [48]. Furthermore, there is a rare means across the BBB called the paracellular aqueous pathway through which small water-soluble molecules diffuse into the brain [49]. 

## 3. Recent Work in BOC Development 

Considering the complex structure of BBB, great efforts have been made to rebuild BBB in vitro. The Transwell-based model is primarily used due to its accessibility, cost-friendliness, and ease of assembly. The Transwell structure is composed of two compartments and a thin membrane. It allows for the co-culture of different kinds of cells and allows for the assessment of the permeability and measurement of TEER [50,51,52,53]. However, the Transwell model is a static system and lacks crucial features such as flow shear stress, which is important to the growth of ECs, tight junction formation, and cell–cell interaction [4,54].

With the advancement of microfabrication and microfluidic techniques, one technique found to be promising is to reconstruct the BBB on a chip. For example, polydimethylsiloxane (PDMS), which is transparent and biocompatible, is commonly used for organ-on-a-chip construction. With soft lithography, it is easy to fabricate microchannels with PDMS and culture cells in the channels [55]. New techniques can overcome the shortcomings of the Transwell model and model the brain to a better extent. Various studies on BOCs have been carried out, which can be classified into the following three types: sandwich models, two-dimensional models, and three-dimensional models.

### 3.1. Sandwich Models

Similar to the Transwell model, the sandwich model also contains three parts: two PDMS layers and a membrane. Two PDMS layers are fabricated by soft lithography. ECs are cultured in the microchannels on one PDMS layer, while brain cells such as astrocytes are cultured in the microchannels on the other layer [39,56,57,58,59,60,61,62,63,64,65,66] (Figure 2).

As early as 2012, Booth et al. [4] made a structure for the coculturing of b. End3 (ECs) with an astrocyte cell line, and used embedded electrodes for real-time TEER measurement, which was considered as a new promising system to characterize the BBB (Figure 2A). Compared with the Transwell model, the sandwich model is a dynamic system that can offer mechanical stimuli such as flow shear stress for enhanced BBB integrity, higher TEER value, and lower permeability [4,56,57,59,62,65].

Given that conventional polymeric membranes are incompatible with in situ high-resolution imaging using optical microscopy [63], efforts have been made to optimize the model structure by using PDMS membranes [14,31] (Figure 2B), polyethylene terephthalate [67], hydrophilized polytetrafluoroethylene nanoporous membranes [68], and nanoporous silicon nitride membranes [63] for better optical imaging, cell–cell interaction, and adhesion. 

PDMS has the advantages of excellent biocompatibility, high transparency, and ease of manufacturing. The polymer network structure of PDMS gives it high permeability relative to other materials, which can be used for cell culture applications as it enables the supply of oxygen and the removal of carbon dioxide [69]. However, the high permeability may also cause some problems when using PDMS. For example, small molecules can diffuse into the bulk polymer, which could result in drug loss and significantly reduce the actual drug delivered, or lead to cross-contamination of adjacent microchannels [70]. Different attempts have been made to overcome this problem. Hirotaka Sasaki et al. [71] reported that the conformal deposition of poly-p-xylylene derivatives (parylenes) on the surface of PDMS substantially suppressed the absorption of Rhodamine B, but it compromised the elasticity and gas permeability of PDMS. B.J. van Meer et al. [72] demonstrated that two commercially available lipophilic coatings (Bay K 8644 and bepridil) were useful for preventing the absorption of small molecules by PDMS. 

Furthermore, pump-free models were developed that could get rid of air bubbles, which were disastrous to ECs [39,73]. This model also allows for media recirculation at perfusion rates similar to physiological conditions without the need for pumps or external tubing (Figure 2C).

### 3.2. Two-Dimensional Models

Although the sandwich model performs much better than the Transwell model in terms of higher TEER and lower permeability, the existence of the membrane in the model hinders direct interaction between ECs and astrocytes. This cell–cell interaction is essential for tight junction formation and BBB integrity [74,75].

To enable the direct interaction between ECs and astrocytes, two-dimensional models have been created, which include a ‘brain’ side and a ‘blood’ side, separated by a group of PDMS microchannels rather than a membrane [74,75,76,77,78,79,80] (Figure 3). There is a popular commercial model for reconstructing BBB, which allows for the co-culture of different kinds of cells and the assessment of other features [76,77] (Figure 3A,B). Furthermore, the two-dimensional model also serves as a high-throughput microfluidic platform for drug screening [80] (Figure 3C).

Although the two-dimensional model overcomes the drawbacks of the sandwich model, there is a large difference in stiffness when compared to the in vivo models. The stiffness of the brain (50 Pa) is much lower than that of in vitro methods (GPa), where plastic and glass are widely used. Hence, the cell behaviors, shape, and other key features are highly affected [7]. In addition, vessels are rectangular in PDMS-based BOCs while the vascular is tubular in the brain. The rectangular shape of microchannels leads to inhomogeneous flow shear stress that affects the shape and behavior of ECs [81] and causes dysregulation of transcription factors that suppress the endothelial response to inflammatory stimuli [82].

Overall, due to the above limitations, it is challenging to rebuild a real vascular structure in three-dimensions with the two-dimensional models.

### 3.3. Three-Dimensional Models

To reproduce the BBB microvascular structure and characterize natural cell–cell interactions, tubular models and self-assembled models in three dimensions have been developed, which are mainly distinguished by the fabrication methods (Figure 4).

One popular method for fabricating tubular models is to place a micro-needle between the inlet and outlet, and then pull the needle out after the collagen has gelled, so a circular microchannel is formed for cell seeding [83,84,85] (Figure 4A).

Herland et al. [86] and F Yu et al. [73] proposed another approach to make tubular microchannels. They used viscous fingering to generate lumens in collagen gels. The microchannel formed in this way is homogenous, loose, and fibrillar, which is closer to the real one in the brain (Figure 4B).

To rebuild the complex brain vascular structure in vitro, a self-assembled model was developed where all cell types spontaneously form a module or tissue that recapitulates the structure of the BBB. A hydrogel was premixed with cells (ECs, astrocytes or neurons) and injected into the microchannels before the ECs self-developed into a vascular network [82,87]. Adriani et al. [87] created a single-layer device with four microchannels, two for 3D hydrogels and two for culture medium (Figure 4C). This allowed for the development of a more complex 3D in vitro model, where other types of brain cells such as pericytes and microglia can be incorporated. However, this model did not allow for the measurement of TEER because it is hard to place electrodes on opposite sides of the endothelium that is surrounded by a solid ECM gel, where an even electrical field cannot be guaranteed due to the device geometry [86]. Several attempts have been made to address this problem [88,89,90]. Partyka et al. [90] used a needle to build two parallel and cylindrical voids within the gel. They measured the current passing across the endothelium by inserting electrodes into the input ports, and obtained TEER values ranging from 1000 to 1200 Ω.cm^2^ (Figure 4D).

Generally speaking, during the past decade, different structures have been utilized to mimic the real brain structure and conditions. Aside from the flow shear stress, cyclic strain can also affect the cell morphology, proliferation, and differentiation [91,92] and enhances the integrity of BBB by promoting tight-junction formation [90,93,94]. In addition, it should be highlighted that rebuilding the chemical environment of the brain can optimize the outcomes. For example, the oxygen level is always neglected in conventional BOCs, which leads to differences between the in vitro and in vivo environments, thus influencing cell metabolism [7]. Nevertheless, as PDMS is known to be oxygen transparent, there is the presence of oxygen for the in vitro models using PDMS. Therefore, it is the level of oxygen that needs to be monitored and controlled to mimic the in vivo environments.

Furthermore, the coculture of different types of cells is another vital strategy to rebuild the real structure of BBB. For example, ECs co-cultured with astrocytes on the BOC can express a higher amount of F-actin than those in a mono-culture system [95]. 

**Figure 4 micromachines-14-00112-f004:**
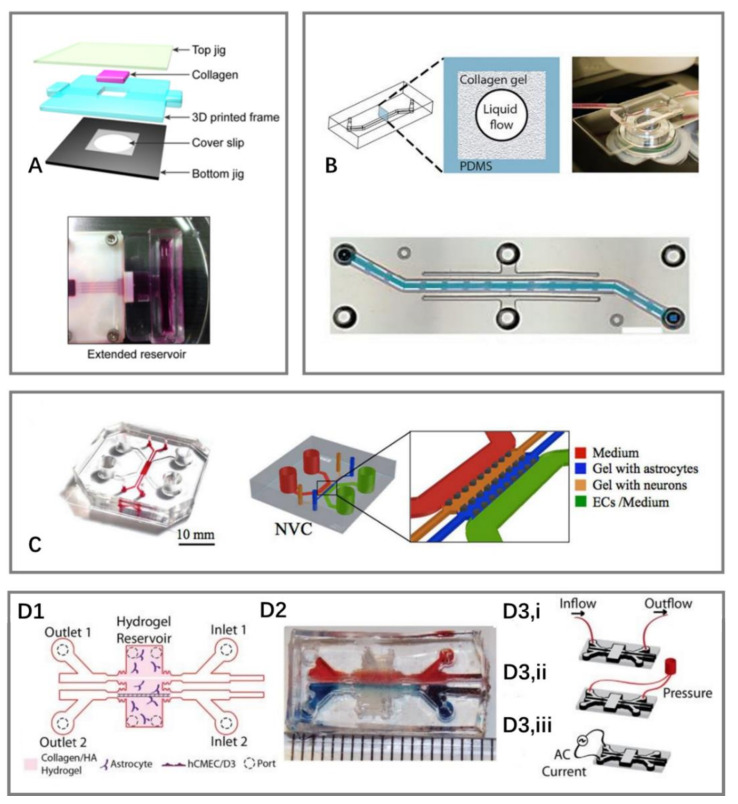
Schematic of different 3D models of BOCs. (**A**) A 3D microfluidic chip for 3D culture, where microchannels were fabricated by microneedles [85]. (**B**) Microchannels fabricated by viscous fingering to generate lumens in collagen gels [86]. (**C**) A 3D self-assembled microvascular microfluidic chip [87]. (**D**) A 3D BBB model that allows for TEER measurement. (**D1**) Schematic of the structure of a microfluidic device. (**D2**) Picture of the device with food dye injected into the two channels. (**D3**) Testing configurations (**i**) application of fluid flow, (**ii**) cyclic strain stimulation, and (**iii**) TEER measurements [90].

### 3.4. Different Means for Evaluating BBB Integrity In Vitro 

Several methods have been developed to evaluate BBB microarrays ex vivo including TEER measurement, protein expression, imaging, and permeability assessment.

The measurement of the TEER value is a ‘gold law’ in evaluating the integrity of the BBB. The higher the TEER value, the better the integrity of the barrier [4,57,60,67]. To test the fully assembled BOC, the electrode wires were connected to an epithelial voltohmmeter through an electrode adaptor (Figure 5A). To calculate the TEER value (TEER), the initial background resistance $R_{b}$ needs to be subtracted from the total resistance $R_{t}$ at each time point and normalized for area A, giving TEER values in Ω cm^2^ [4,57]. The TEER value is calculated with the following equation:$$TEER=\left(R_{t}-R_{b}\right)A$$

Notably, the TEER value of a dynamic system is much higher than a static one (Transwell model) because of the mechanical stimulations. The TEER value of a co-culture model is also much higher than a mono-culture model, as previously mentioned [4,65]. 

It has also been suggested that the mRNA levels of INSR, VE-cadherin, MRP1, and ZO-1 in the iPS-derived microvascular endothelium are significantly increased when ECs are co-cultured with astrocytes and pericytes [4,65]. Therefore, by detecting the expression of proteins such as ZO-1 and claudin-5, the integrity of the BBB can be assessed by immunofluorescence microscopy [4,31,65,67] (Figure 5B). Furthermore, transmission electron microscopy and confocal microscopy have been employed to directly examine the opening of tight junctions after treatment [21,60,65] (Figure 5C,D).

The permeability assessment is another important means to evaluate the BBB integrity as the barrier hinders the transport of substances into the neural compartment. Different kinds of dextrans have been used as fluorescent tracers to analyze the permeability of the endothelial barrier [4,31,65,74]. The diffusion rate of tracer molecules across the BBB membrane was analyzed based on the fluorescence intensity in the microchannel (Figure 5).

The equation for calculating the permeability (P) is [95]:$$P=\frac{1}{I_{vo}}\cdot \frac{V}{S}\;\frac{dI_{t}}{dt}$$
where

$I_{t}$ = the average intensity of dextran in the tissue compartment at time point *t*;

$I_{vo}$ = the maximum fluorescence intensity of the vascular channel;

$\frac{V}{S}$ = the ratio of volume to surface area of the vascular channel.

## 4. Ultrasound-Driven Microbubbles Reversibly Open BBB 

A sound wave with a frequency higher than 20 kHz is called ultrasound, which is widely used in numerous applications, particularly as an imaging tool for clinical diagnosis [96]. Microbubbles are frequently utilized as ultrasound contrast agents (UCA). These bubbles can be generated by mixing water and gas. They range in size from 1–10 μm, and are encapsulated by a lipid or a protein shell [97]. Apart from their utilization as UCA, microbubbles also serve as therapeutic agents [98]. With the advancements in research, ultrasound-driven microbubbles have been found to generate bioeffects that can improve the permeability of the BBB and promote drug uptake [96]. 

Due to the high efficiency and low toxicity for local drug delivery, ultrasound-driven microbubbles have demonstrated good potential for delivering drugs to the brain [99]. Specifically, low acoustic pressure enhances drug uptake by primarily stimulating endocytosis, while high acoustic pressure results in drug uptake by inducing membrane pores [100,101].

### 4.1. Oscillation and Cavitation of Microbubbles Driven by Ultrasound 

Microbubbles can form stable cavitation and inertial cavitation when exposed to ultrasound, which could induce different biological effects (Figure 6). The state depends on the sound pressure. At low driven acoustic pressure, microbubbles will expand and shrink around a balanced radius. It has been reported that low acoustic pressure caused mild cell membrane deformation, leading to upregulated endocytosis [100]. At high driven acoustic pressure, microbubbles will experience violent expansion and collapse [102]. This may create jetting and shear flow [103,104,105,106], which pushes microbubbles toward ECs and creates cell membrane pores (sonoporation) or cell mortality [100]. 

Aside from sonoporation, the generated mechanical force also disrupts the cell tight junctions and creates gaps between cells. It should be highlighted that this disruption is reported to be reversible after several hours both in vivo and in vitro [21,107].

In general, the demonstrated results show that the cell membrane pores or undermined cell–cell contact caused by ultrasound and microbubbles can promote drug delivery. For more information on the bio-effects induced by ultrasound and microbubbles, please refer to previous reviews [97,108,109].

### 4.2. BBB Opening by Ultrasound-Driven Microbubbles In Vivo and In Vitro 

#### 4.2.1. In Vivo Experiments

There is a long history of ultrasound and microbubbles being used to open the BBB in small animals [13,110,111,112], large animal models [12,113,114,115], or humans [116,117,118] for drug screening or mechanistic study.

Chen at al. [13] demonstrated that magnetic resonance imaging (MRI)-guided focused ultrasound enhanced the delivery and time-dependent diffusion of radiolabeled nanoclusters (^64^Cu-CuNCs) in the brain tumor of a diffuse intrinsic pontine glioma mouse model (Figure 7A). Marquet et al. [113] first demonstrated the feasibility of transcranial microbubble-enhanced, focused ultrasound BBB opening and subsequent BBB recovery in non-human primates. (Figure 7B). During a clinical trial, Carmen et al. [116] transiently disrupted the BBB in human subjects with Parkinson’s disease with MRI-guided focused ultrasound. The safety, feasibility, and reversibility of ultrasound mediated BBB opening were demonstrated in this study (Figure 7C).

Though in vivo models can yield accurate therapeutic results, the drawbacks of the high cost and time consumption cannot be ignored, not to mention the ethical problems. Furthermore, it is not easy to perform quantitative or parametric studies with in vivo models. 

Given the above reasons, in vitro models are largely used instead, which can be assorted into static systems and dynamic systems.

#### 4.2.2. In Vitro Static System

In static systems, ECs are cultured in the Transwell model [119,120,121]. Cell monolayer culture [100,122] and cell suspensions [123,124] are easy to obtain and assemble. Ohta et al. [121] mimicked the BBB by culturing mouse bEND.3 (ECs) in a Transwell model and investigated the optimal size of nanoparticles that could be delivered into the brain by ultrasound and microbubble stimulation. With the exposure of focused ultrasound, the permeation of 3- and 15-nm gold nanoparticles (AuNPs) across such an in vitro BBB model was significantly increased by up to 9.5 times, and the smaller diameter of the nanoparticles means a higher permeability across the BBB (Figure 8A). Furthermore, Beekers et al. [122] cultured a primary HUVEC monolayer in an acoustic compatible cell culture chamber. They reported that ultrasound could reversibly disrupt the BBB, and the opening of cell–cell contact is a biological response as a consequence of sonoporation, instead of an independent drug delivery pathway (Figure 8B).

However, it should be emphasized that the static system is oversimplified and lacks some crucial features such as shear stress and interactions between different kinds of cells, which largely affects the BBB integrity, as previously mentioned.

#### 4.2.3. In Vitro Dynamic System

To overcome the above drawbacks, microfabricated chips were used to study the disruption of endothelial cell–cell contract (Figure 9, Figure 10, Figure 11 and Figure 12) by ultrasound and microbubbles [21,37,101,125,126]. A typical setup for this type of experiment is illustrated in Figure 13. 

Silvani et al. [21] used a 2D commercial model to reconstruct the microvascular in vitro and stimulated the modeled endothelial barrier by ultrasound and microbubbles. They found that a higher acoustic pressure (0.72 MPa) and lower flow rate (0.5 µL/min) led to a bigger gap number and a larger gap area of cell–cell contact compared to low acoustic pressure (0.4 MPa) and a high flow rate (25 µL/min) (Figure 9). It was also observed that after ultrasound exposure, the endothelial barrier reverted to the original state in a short time (45 min) (Figure 9).

Schreur [127] designed a sandwich model to mimic the BBB in vitro and to compare different types of ECs. It was reported that human cerebral microvascular endothelial cells/D3 (hCMEC/D3) had a higher ZO-1 expression and TEER value compared to HUVEC and hiPSC-BMEC. This shows that hCMEC/D3 co-cultured with astrocytes have a higher ZO-1 expression but lower TEER value. After ultrasound and microbubble treatment, the TEER value decreased first and then recovered (Figure 10). 

Royse et al. [126] reported a perfusion-based 3D printable hydrogel vascular model to assess BBB permeability and its function. Ultrasound-driven microbubbles increased the permeability of the BBB, altered cell area, and disrupted cell–cell junctions. However, this model did not demonstrate the physiologic recovery of cells after ultrasound exposure (Figure 11A1,A2).

DeOre et al. [125] reconstructed a 3D microvascular structure through a previously reported 3D model [128], where the channel was created by a needle. They found that the flow rate was an important parameter. A higher intensity of sonication (1 W/cm^2^), and a low flow rate (10 µL/min) can result in a greater reduction in both the cell area and junctional integrity as well as disrupting actin cytoskeleton (Figure 11B1–B3).

Park et al. [101] modeled a 3D microvascular structure in a microfluidic chip. They found that the perfusion of microbubbles can significantly enhance the endocytosis of drugs at a low intensity of ultrasound treatment, and minimal cellular damage was observed for both the microbubbles and untargeted doxorubicin-encapsulating liposomes (DOX-liposomes) perfused through microvessels in the chip (Figure 12A1–A3).

Juang et al. [37] presented a perfusable in vitro model for the 3D culture of ECs. They found that higher HUVEC permeabilization efficiency, enhanced drug delivery, and greater cell death occurred at a larger ultrasound amplitude (Figure 12B1–B3). In addition, in the presence of ultrasound, clusters of large bubbles could form beneath the microbubbles infused to the channel. These bubble clusters were observed to push the infused microbubbles against the vessel wall and could improve the treatment efficiency. 

**Figure 9 micromachines-14-00112-f009:**
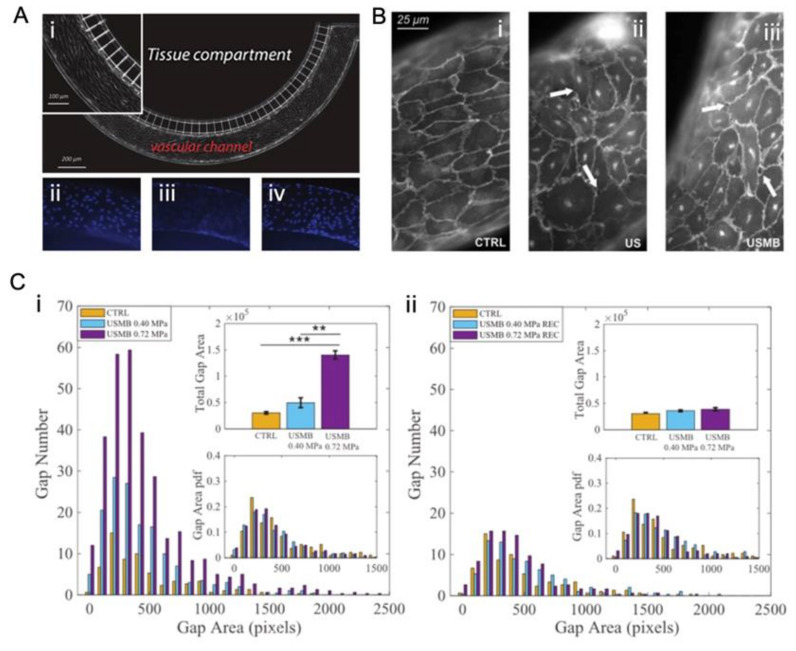
A 2D in vitro dynamic system for the study of ultrasound mediated opening of vascular barriers. (**A**) The model is composed of a tissue compartment and vascular channel [21]. (**i**) Brightfield image of HUVECs seeded in the vascular channel. The inset highlights the elongation of cells in the direction of flow. (**ii**–**iv**) Fluorescence images of cell nuclei after DAPI staining. (**ii**,**iv**) shows the nuclei of cells adhered on the top and bottom side of the channel, respectively. (**iii**) focuses on the nuclei of cells adhered on the lateral walls of the channel. (**B**) Confocal fluorescence images of VE-cadherin showing a region of confluent endothelium cultured at 25 µL/min fluid flow. (**i**) Untreated sample (CTRL). (**ii**) The sample exposed only to ultrasound (US). (**iii**) The sample exposed to ultrasound driven microbubbles (USMBs) with an acoustic pressure of 0.72 MPa, corresponding to a piezo driving voltage of 140 mV. US exposure protocol: duration 30 s, 500 cycles, frequency 1 MHz. Arrows highlight typical intercellular openings [21]. (**C**) Reversible opening of cell junctions by ultrasound-driven microbubbles in an artificial endothelial layer. Gap number and gap area (**i**) right after ultrasound irradiation and (**ii**) 45 min after ultrasound exposure [21].

**Figure 10 micromachines-14-00112-f010:**
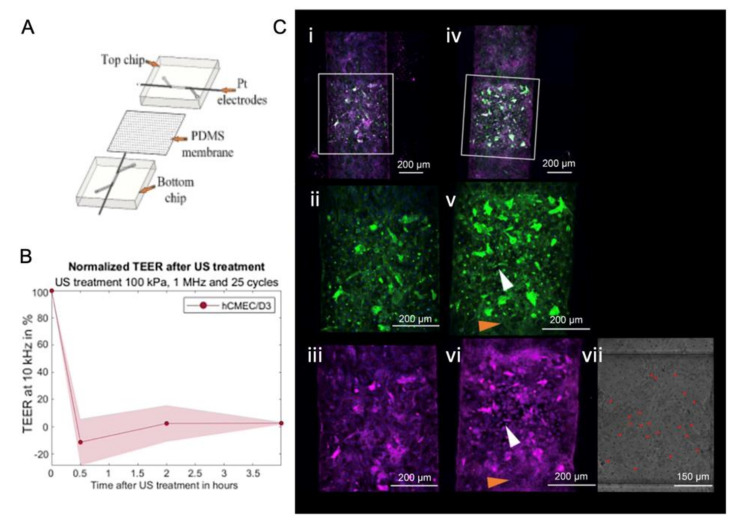
A sandwich model that mimics the BBB in vitro was employed for the study of the ultrasound mediated opening of vascular barriers. (**A**) The structure and components of the model. (**B**) Normalized TEER value changed after focused ultrasound and microbubble treatment (FUS + MB) [127]. (**C**) Channels with a monolayer of hCMEC/D3 exposed to ultrasound (**i–iii**) without microbubbles, (**iv**–**vi**) with microbubbles. FUS + MB treatment of 25 cycles, 1 MHz, and 100 kPa, the red dots in (**vii**) denote microbubbles, white arrows in (**v**,**vi**) designate the same regions of cells that responded to FUS + MB treatment, with ZO-I stained in pink, nuclei stained in blue, and F-actin stained in green [127].

**Figure 11 micromachines-14-00112-f011:**
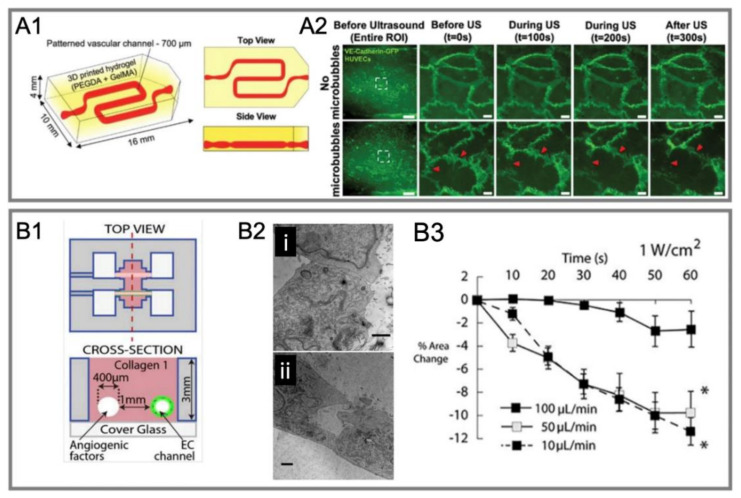
Recent studies on ultrasound-mediated opening of vascular barriers in three-dimensional BOC. (**A1**) A perfusion-based 3D printable hydrogel vascular model for culturing HUVEC [126]. (**A2**) Live imaging of endothelial cell–cell junctions under ultrasound exposure [126]. (**B1**) A 3D model where channels were fabricated by micro-needles [125]. (**B2**) Transmission electron microscopy for testing conditions of (**i**) 0.5 W/cm^2^ ultrasound intensity and 100 μL/min fluid flow and (**ii**) 1 W/cm^2^ ultrasound intensity and 10 μL/min fluid flow. (**B3**) The influence of different flow rates on the cell area reduction under acoustic intensity of 1 W/cm^2^ [125].

**Figure 12 micromachines-14-00112-f012:**
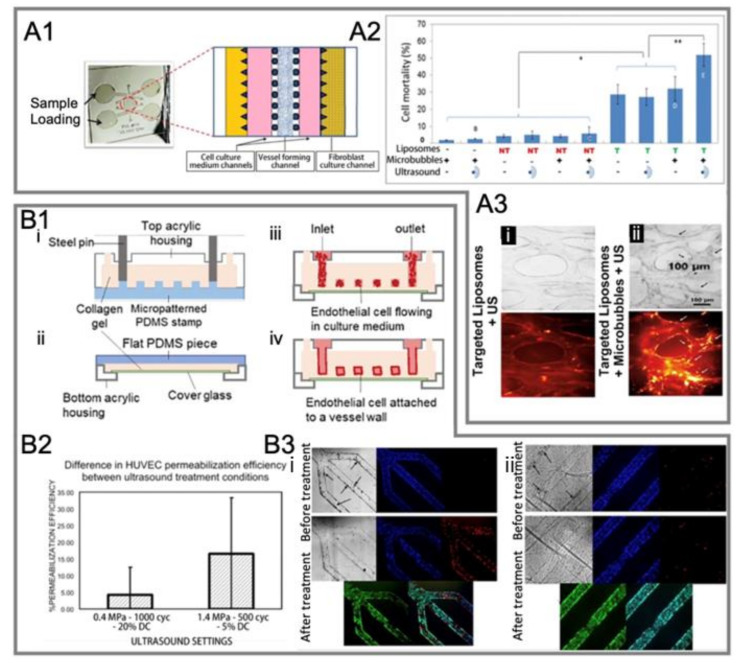
Another two representative studies on the ultrasound-mediated opening of vascular barriers in three-dimensional BOC. (**A1**) A microvascular chip comprises several channels partitioned by micro-posts [101]. (**A2**) Effect of different test conditions on cell mortality in the microvasculature [101]. (**A3**) Microbubbles increased the toxicity of DOX-loaded liposomes to cells under ultrasound exposure [101]. Microvessels were perfused with (**i**) integrin-targeted liposomes alone or (**ii**) integrin-targeted liposomes + microbubbles, then washed and treated with ultrasound. Bright field (**top**) and TRITC fluorescent (**bottom**) images were taken after 24 h. Black arrows in the bright-field image in panel (**ii**) show cells extruded from nearby vessels. White arrows in TRITC image in panel (**ii**) show liposomes that have penetrated the external vessel walls. (**B1**) A perusable in vitro model for the 3D culture of ECs [37]. (**i**,**ii**) Preparation of the microvascular network (MVN). (**i**) The top half of the MVN: liquid collagen I was injected into the space enclosed by a patterned PDMS stamp and a top acrylic housing. (**ii**) The bottom half of an MVN: a thin collagen layer was applied on top of a cover glass that sits above the window of a bottom acrylic housing. (**iii**) After combing the two MVN halves, HUVECs were introduced to the collagen scaffold. (**iv**) HUVECs attached to the collagen walls overnight and formed a vascular endothelium after 2–3 days of culture. (**B2**) Assessment of HUVEC permeabilization efficiency under different ultrasound treatment conditions [37]. (**B3**) Bright field and fluorescence images of the vascular barrier before and after ultrasound-microbubble treatment under acoustic conditions of (**i**) 1.4 MPa, 500 cycles, 5% duty cycles (DC) and (**ii**) 0.4 MPa, 1000 cycles, 20% DC. Hoechst (blue) stains HUVEC nuclei; PI (red) stains damaged plasma membranes of cells, which indicates sonoporation or cell death [37].

**Figure 13 micromachines-14-00112-f013:**
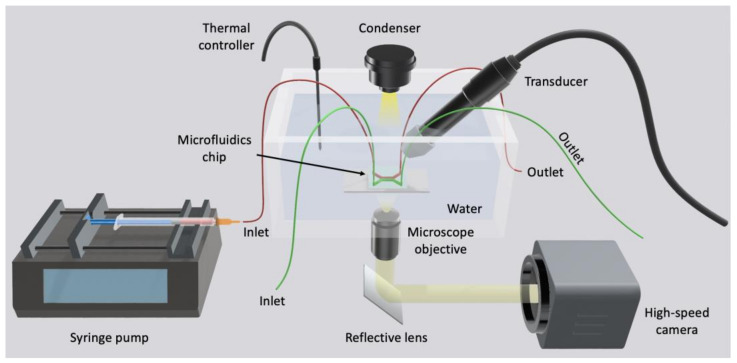
Schematic of a typical setup used for the ultrasound experiments with BOCs. The chips were placed in the water bath. Ultrasound was generated by the transducer placed at a 45° angle to the chip surface. A stroboscopic light source illuminated the chip while the high-speed camera captured the MB dynamics. The syringe pump perfused microbubbles into the chip.

Different ultrasound parameters were used in these studies including ultrasound frequency, acoustic pressure/intensity, pulse duration, and duty cycle (see Table 2). It is worth noting that the flow rate in the BOC also affects the disruption of the vascular barrier.

## 5. Conclusions and Future Prospects

In this review, we discussed the recent works on various designs of BOCs such as sandwich models, 2D models, and 3D models. We also introduced recent advances in opening the endothelial barrier by ultrasound-driven microbubbles. Although enormous progress has been made in the past decade, it should be noted that there are still areas that can be improved.

By changing the structure and experimental conditions, the BBB can be modeled better in vitro. Chasing a model with a similar size to the real BBB, Marino et al. [129], for the first time, presented a 1:1 scale, biomimetic, and biohybrid BBB model through two-photon lithography. The model is characterized by a vessel diameter close to the size of microcapillary and fluid flow, similar to the physiological situation in the body. Aside from the structure, another improvement is to provide and mimic the mechanical stimuli. However, besides shear stress, blood flow also causes cyclic strain. Partyka et al. [90] found that the cyclic strain affected ECs in a microfluidic platform. In addition, BBB models based on human-induced pluripotent stem cells are attractive, and these platforms offer a promising avenue for personalized medicine [79,130,131]. In summary, better in vitro models of the BBB can be obtained by closely mimicking the real structure, providing more mechanical stimuli that is physiologically relevant and using human induced pluripotent stem cells. Moreover, it is important to recreate the chemical environment such as the oxygen level, as this can lead to the differences between in vitro and in vivo models and influence cell metabolism [7].

Once established and improved, the BOC may provide an easy-accessible platform that can generate more experimental data for computational modeling. The number of in vivo studies for computational modeling is limited and the data are imbalanced, hindering the development of more effective computational prediction [132]. The high cost and long test period of animal models are major barriers for data collection, which can be alleviated by the BOC. 

To optimize the results of the sonoporation of BBB, it should be emphasized that for disruption of the endothelial barrier by ultrasound-driven microbubbles, as discussed in this review, we used a microvascular model. In other words, future studies could be improved by co-culturing different types of cells to reconstruct the BBB model. Moreover, as previously mentioned, previous studies have focused on evaluating the effects of acoustic pressure/intensity and ultrasound frequency while the influence of other parameters may not be well-considered. The single pulse duration, pulse repetition frequency/duty cycle, and flow rate remain to be explored. Furthermore, as we discussed in this review, most of the recent studies on ultrasound-driven microbubbles interacting with the endothelial barrier have investigated BBB changes by altered TEER values, protein expression, gap junction opening, and altered permeability. However, how gap junctions are opened and restored is unclear. Further studies of the dynamic process of BBB opening and the underlying mechanism is necessary. One possible method is to track the cellular signaling through calcium imaging and monitor the morphology change of the tight junctions. Investigating the kinetic process and mechanism of BBB opening by ultrasound will contribute to both the fundamental understanding and development of new strategies for drug delivery to the brain. 

## Figures and Tables

**Figure 2 micromachines-14-00112-f002:**
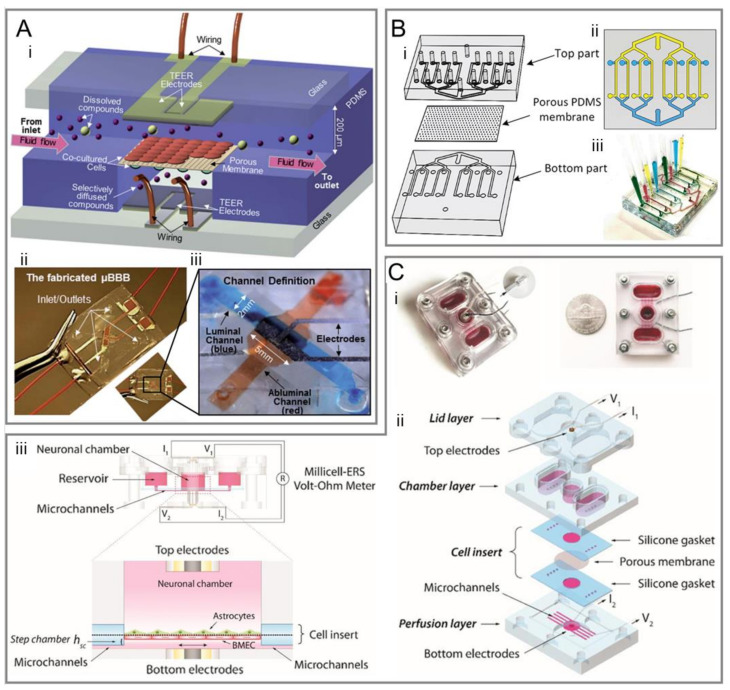
Representative sandwich models of BOCs. (**A-i**) Schematic of a popular sandwich design comprises two perpendicular flow channels and electrodes for TEER measurement. (**A-ii**) The assembled μBBB chip as shown in panel (**i**). (**A-iii**) An enlarged view of the perpendicular flow channels and electrodes [4]. (**B-i**) Schematic of a multiplexed chip with eight parallel channels separated by a porous PDMS membrane. (**B-ii**) Top view of the two-layer chip design. (**B-iii**) The result of creating eight different conditions in the fabricated two-layer chip [31]. (**C-i**) Fabricated device of a pump-free microfluidic chip. Red dye was used to visualize the microchannels, neuronal compartments, and reservoirs. (**C-ii**) Schematic for the components of the device: a bottom perfusion layer with microchannels and bottom electrodes; a middle chamber layer that forms reservoirs and the neuronal chamber; a top lid layer with top electrodes, which covers the neuronal chamber and the reservoirs to minimize fluid evaporation; a cell insert part made from two silicone sheets and a sandwiched porous polycarbonate membrane. The cell insert part was assembled between the bottom and the middle layers. (**C-iii**) Side view of the device structure, including fluid chamber, microchannels, electrodes connected to a Millicell-ERS Volt-Ohm Meter and the co-cultural orientation of BBB cells [39].

**Figure 3 micromachines-14-00112-f003:**
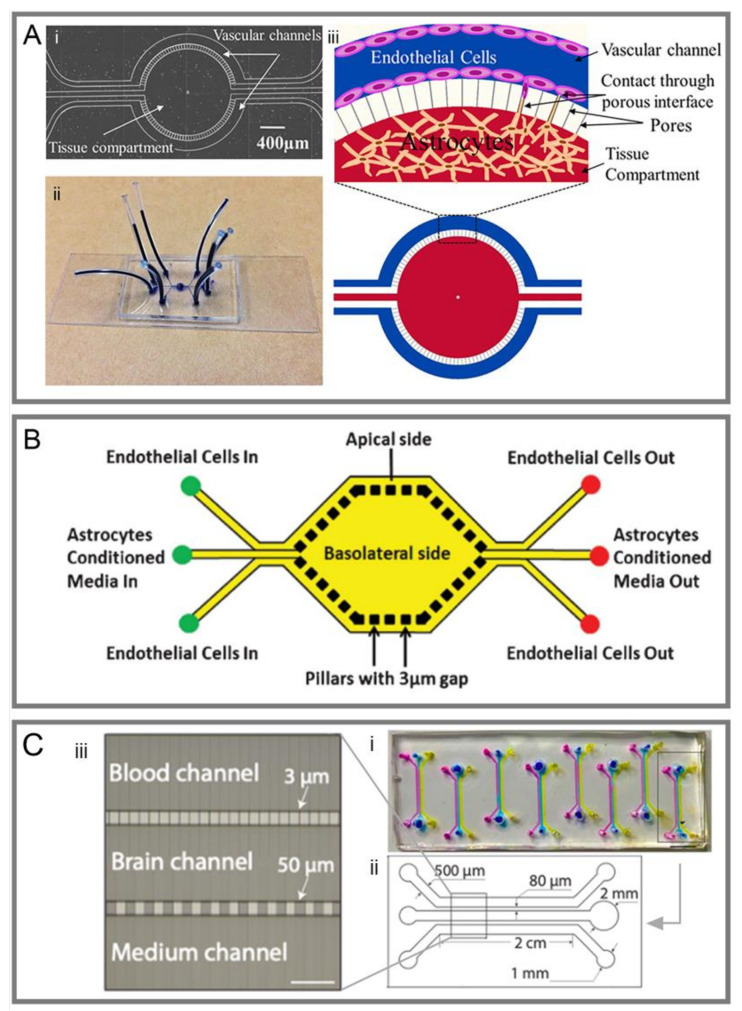
Schematic of different 2D models of BOCs. (**A-i**) The mask design of a microfluidic chip consisting of circular vascular channels and tissue compartment. (**A-ii**) The fabricated BOC on a microscope glass slide with plastic tubing and individual vascular channels and the tissue compartment stained (dark blue). (**A-iii**) Schematic illustration of cell culture in the BOC showing the endothelial cell lining on the walls of the vascular channel (blue), astrocytes contained in the tissue compartment (red), and their contact through the porous interface [74]. (**B**) Another design of 2D BOC with the apical and basolateral sides separated by 3 μm gaps formed by microfabricated pillars. Apical side contains endothelial cells while the basolateral side contains astrocyte conditioned media [75]. (**C**) A 2D microfluidic device for high-throughput drug assessment. (**i**) Top view of the fabricated microfluidic chip, which contains eight independent units; (**ii**) Each unit consists of three main channels. For each channel, the width and height is 500 μm and 100 μm, respectively, and the length for the parallel part is 2 cm. (**iii**) An enlarged view of the channel structure in the parallel part. An array of microchannels (3 μm in width) is in between the brain channel and the blood channel. Another array of microchannels (50 μm in width) is in between the brain and the media channels [80].

**Figure 5 micromachines-14-00112-f005:**
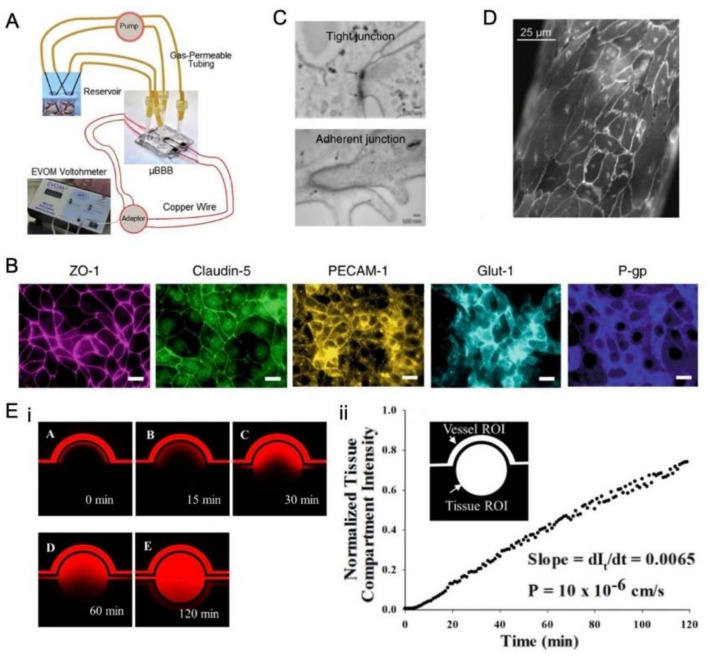
Different means to evaluate the BBB integrity. (**A**) TEER measurement in a BOC device, where copper electrode wiring is connected to the voltohmmeter via an electrode adapter [4]. (**B**) Immunofluorescence micrographs of the human brain endothelium cultured on-chip for 3 days, showing high expression levels of ZO-1, claudin-5, PECAM-1, GLUT-1, and P-glycoprotein (bar, 20 µm) [65]. (**C**) Electron micrograph of human brain microvascular endothelium after 3 days of culture in the BBB chip. The presence of well-formed tight junctions (top, pointed by arrows) and adherent junctions (bottom, indicated by arrows) are highlighted [65]. (**D**) Confocal fluorescence imaging of VE-cadherin for a portion of the endothelium after US exposure with the presence of MBs [21]. (**E-i**) Permeability of Texas Red dextran (40 kDa) from the vascular channel to the tissue compartment in the cell-free BOC after 5 min, 15 min, 30 min, 60 min. and 120 min from the start of flow in the vascular channel. (**E-ii**) Quantification of the normalized intensity of the dextran in the tissue compartment as shown in (**E-i**). The intensity increased linearly with time [74].

**Figure 6 micromachines-14-00112-f006:**
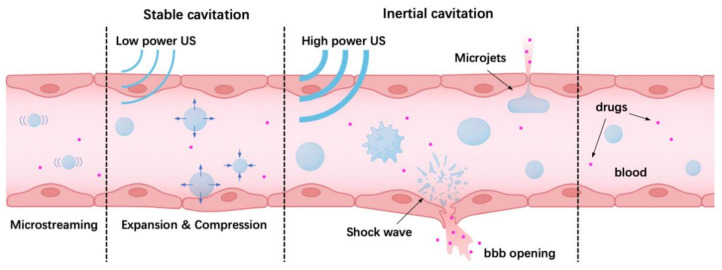
Microbubbles exhibit two states (stable cavitation and inertial cavitation) under different acoustic pressures, leading to different bioeffects.

**Figure 7 micromachines-14-00112-f007:**
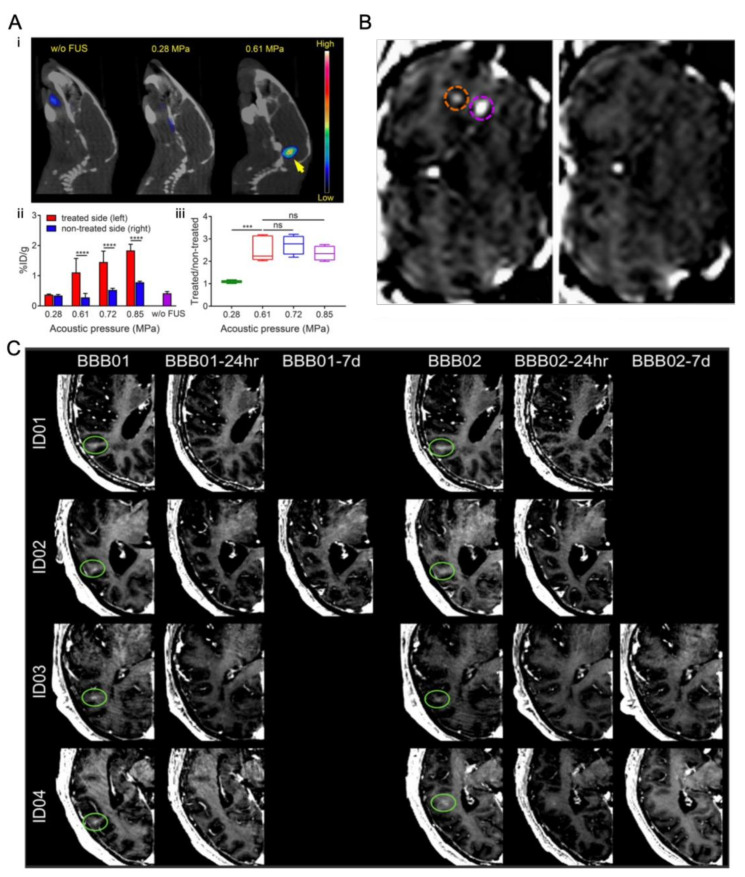
Representative studies for BBB opening by ultrasound-driven microbubbles in vivo. (**A**) (**i**) In vivo PET/CT images of ^64^Cu-CuNCs without focused ultrasound (FUS) treatment and under 0.28 and 0.61 MPa FUS pressure in WT mice at 24 h post intravenous injection. (**ii**) Quantitative analysis of ^64^Cu-CuNC uptake in FUS-treated left pons and nontreated right pons of WT mice under 0.28, 0.61, 0.72, and 0.85 MPa FUS pressures as well as in the pons without FUS treatment in WT mice. (**iii**) Uptake ratios of the treated sites vs. nontreated sites of the FUS-treated mice under different pressures (*** *p* < 0.001, **** *p* < 0.0001, n = 4−5) [13]. (**B**) Coronal MR image after intravenous injection of gadodiamide 1 h after sonication, showing the presence of gadodiamide in the brain parenchyma confirming the local disruption of BBB. Two opening sites under different ultrasound treatment are circled (purple circle 0.3 MPa and orange circle 0.45 MPa). 6 days after sonication, coronal MR image confirming that the BBB is closed and the procedure is reversible [113]. (**C**) Brain CT images (gadolinium enhancement in T1-weighted) of patients 1–4 immediately after the BBB opening procedure (BBB01). The BBB opening was closed after 24 h (stage 1) in patients 1, 3, and 4 (BBB01-24 h) and in patient 2 on the seventh day of MRI follow-up (BBB01-7d). For stage 2 treatment, the BBB was closed in patients 1 and 2 after 24 h and in patients 3 and 4 in the following MRI study on day 7 (BBB02-7d) [116].

**Figure 8 micromachines-14-00112-f008:**
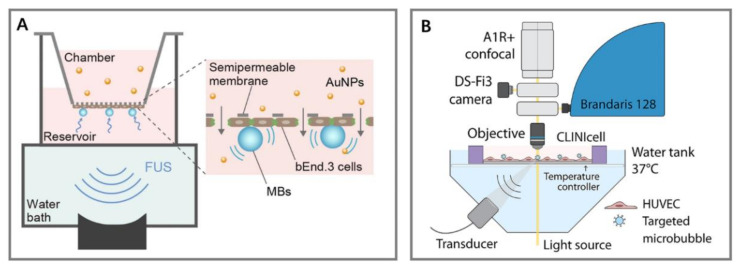
Different kinds of in vitro static systems employed for the study of ultrasound mediated opening of the BBB. (**A**) Ultrasound-driven microbubbles stimulate ECs in the Transwell model [120]. (**B**) Ultrasound stimulates HUVEC in the monolayer culture model [122].

**Table 2 micromachines-14-00112-t002:** A list of the ultrasound parameters used for the disruption of the endothelial barrier by ultrasound-driven microbubbles in recent studies.

Frequency	Acoustic Pressure /Intensity	Pulse Duration	Duty Cycle	Flow Rate	Reference
1 MHz	0.1 MPa or 0.5MPa	25 μs	Not mentioned	Not mentioned	[127]
1 MHz	0.4 MPa	1 ms	20 %	10 µL /min	[37]
1 MHz	1.4 MPa	500 μs	5%		
1 MHz	2.0 W/cm^2^	Not mentioned	50 %	100 µL /min	[126]
1 MHz	0.4 MPa	500 μs	0.1 %	0.5 µL/min	[21]
	0.72 MPa			25 µL/min	
1.1 MHz	0.8 MPa	10 μs	0.1 %	Not mentioned	[101]
3 MHz	0.5 w/cm^2^	N.A.	Continuous	10, 50, and 100 μL/min	[125]
	1 w/cm^2^	N.A.	Continuous		

## Data Availability

Not applicable.

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
