# Peer review of "Advances in BBB on Chip and Application for Studying Reversible Opening of Blood–Brain Barrier by Sonoporation"

_micromachines, 2022, doi:10.3390/mi14010112_

Round 1

Reviewer 1 Report

The manuscript presents a rather concise review on BBB in vitro models and the use of sonoporation to transiently open the BBB and deliver drugs through it.

The topic is timely and of interest to a potentially wide audience, however the devices are often described only too summarily or at high level to justify inclusion in this journal.

Formatting and in particular the English grammar, wording and typos need a thorough revision. In present form the language is not sufficient to be granted publication. Some of the captions are too short and not enough descriptive of the figures (caption for figure 1, 2, 3). Table 1 is too large for the page, same for Figure 7.

On page 5 and later, the authors mention the use of PDMS for the BoC models. PDMS is known to be oxygen transparent, which should be related to the oxygen presence in the in vitro models discussed later (page 8), and particularly suffers from absorption of small hydrophobic molecules including drugs. This can have an influence on the drug screening assays, and the authors should discuss this aspect and alternative materials (partly done only for membranes at the bottom of page 5).

The authors write that cyclic strain is important for the BBB model, however support this with only a single reference [88]. More evidence is needed to support this claim.

Section 3.4 on methods to evaluate BBB integrity is rather shallow and extremely poor in references. Similar observations for Section 4.2.1 on in vivo experiments.

Several statements throughout the manuscript are not well written or wrong. For instance: in the Abstract, it says that the BBB hinders drug development, which is too general of a statement; at the end of the Abstract, what in vivo platform is meant? On page 5 and 6, "free pump" --> "pump-free"; in the caption of figure 4, what does "self-developed" mean? The title of Section 4 is not clear. In page 15, what is meant by "brain-size models"? Taken literally, that would not qualify as an organ-on-chip.

In the last section, the authors comment that many parameters of ultrasound are not examined yet. It will help that the authors introduce a table with parameters of ultrasound stimulation used for BBB disruption, and their values used in the most relevant models.

Author Response

Thank you for your comment, the details of the response can be seen in the document

Reviewer 2 Report

The presence of BBB hinders drug delivery to the brain for treating Alzheimer’s disease and other dementias. There is an increasing demand for a new in vitro platform that can be used for BBB drug delivery studies. In this review, recent advances and developments of BOC (BBB on chips) are well presented. The authors further highlighted the emerging research areas for ultrasound-induced BBB opening on BOC. The advantages and drawbacks of these prior studies are also discussed, which may inspire better BOC designs and optimizing the experimental design for ultrasound mediated reversible BBB opening. The topic is important and the manuscript is well organized. However, for integrity, a more detailed and updated review about the advances and challenges for ultrasound mediated BBB opening for in vivo experiment including animals and human trial are necessary.

Besides, there are typo errors and mistakes, e.g.,

1. At line 23, in vivo should be changed to in vitro;

2. At line 150-151,   ‘which is important to ECs the growth for tight junction formation, and cell-cell interaction’, does it mean ‘which is important for the growth of ECs, tight junction formation, and cell-cell interaction’?

3. At line 270 ‘Ultrasound-driven microbubbles reversibly opening disrupts BBB’, opening and disrupts are repeated.

Author Response

Thank you for your comment, the details of the responses can be seen in the document.

Round 2

Reviewer 1 Report

The authors proposed a thorough revision of the original manuscript in view of the received feedback. The formatting of the large pictures remains an issue, as well as the positioning of Table 2 at the very end of the paper, which is unusual. Also not clear what "BBB phenomenon" in caption of figure 1 means.

Author Response

More details can be ssen in the document
